# Microphysiological Models for Mechanistic-Based Prediction of Idiosyncratic DILI

**DOI:** 10.3390/cells12111476

**Published:** 2023-05-25

**Authors:** Sydney Stern, Hongbing Wang, Nakissa Sadrieh

**Affiliations:** 1Department of Pharmaceutical Sciences, University of Maryland School of Pharmacy, 20 Penn Street, Baltimore, MD 21201, USA; sydney.stern@rx.umaryland.edu; 2Office of New Drugs, Center of Drug Evaluation and Research, FDA, 10903 New Hampshire Ave, Silver Spring, MD 20993, USA

**Keywords:** drug-induced liver injury, idiosyncratic, microphysiological models, hepatotoxicity

## Abstract

Drug-induced liver injury (DILI) is a major contributor to high attrition rates among candidate and market drugs and a key regulatory, industry, and global health concern. While acute and dose-dependent DILI, namely, intrinsic DILI, is predictable and often reproducible in preclinical models, the nature of idiosyncratic DILI (iDILI) limits its mechanistic understanding due to the complex disease pathogenesis, and recapitulation using in vitro and in vivo models is extremely challenging. However, hepatic inflammation is a key feature of iDILI primarily orchestrated by the innate and adaptive immune system. This review summarizes the in vitro co-culture models that exploit the role of the immune system to investigate iDILI. Particularly, this review focuses on advancements in human-based 3D multicellular models attempting to supplement in vivo models that often lack predictability and display interspecies variations. Exploiting the immune-mediated mechanisms of iDILI, the inclusion of non-parenchymal cells in these hepatoxicity models, namely, Kupffer cells, stellate cells, dendritic cells, and liver sinusoidal endothelial cells, introduces heterotypic cell–cell interactions and mimics the hepatic microenvironment. Additionally, drugs recalled from the market in the US between 1996–2010 that were studies in these various models highlight the necessity for further harmonization and comparison of model characteristics. Challenges regarding disease-related endpoints, mimicking 3D architecture with different cell–cell contact, cell source, and the underlying multi-cellular and multi-stage mechanisms are described. It is our belief that progressing our understanding of the underlying pathogenesis of iDILI will provide mechanistic clues and a method for drug safety screening to better predict liver injury in clinical trials and post-marketing.

## 1. Introduction

Drug-induced liver injury (DILI) is an adverse drug reaction that is a major concern for clinicians, pharmaceutical companies, and regulatory agencies, leading to the withdrawal of drugs from the market [1]. According to the United States Food and Drug Administration (FDA), DILI describes hepatotoxicity caused by hepatocellular injury, indicated by the leakage of aminotransferase (AT) enzymes from injured liver cells without prominent evidence of hepatobiliary obstruction or intrahepatic cholestasis, and the pattern of injury can vary [2]. While rare (4–19 per 100,000 people in the population and 30–33 per 100,000 people in the healthcare system) in general, it is the most common cause of acute liver failure in the US and Europe [3,4,5]. Nine percent of patients impacted by DILI either die or require liver transplantation and 19% show chronic liver damage 6 months after DILI onset [6]. Among over 1100 drugs causing DILI, 13% of acute liver failures are considered idiosyncratic and independent of dose, route of administration, and duration [7,8]. Although the complex mechanism of DILI remains unclear, both immune-mediated and non-immune-mediated mechanisms, drug–drug interactions, drug properties, genetic variations in drug-metabolism enzymes (DMEs) and transporters, and other host factors have been investigated.

Before human trials occur, drugs are evaluated in preclinical models including both animal models and in vitro systems. The evidence gathered using animal models, however, sometimes provides insufficient prediction of drug hepatotoxicity in humans. Many drugs that eventually show clinical hepatotoxicity were not prevented from proceeding into clinical trials by findings in animal studies. Additionally, treatment of animals with drugs that are known to cause human DILI do not elicit the same results as those seen in humans, even at concentrations that are markedly higher [9,10,11]. In animal models that do recapitulate liver injury, the phenotype often differs from that of humans, potentially depicting differences in the underlying mechanism [12,13,14]. In Addition to the species variation highlighted, there is an ethical motivation to reduce the use of animals in research. Thus, there is a significant interest and unmet need in the development of novel in vitro models that complement and enhance in vivo animal data to predict DILI.

Research thus far has largely focused on DILI that occurs at toxic doses of a compound, often exhibits dose-dependency, and is therefore predictive, with acetaminophen, the most commonly reported drug associated with DILI, as a prototypical model compound [15,16]. While improved in vitro microphysiological systems hold the potential to predict iDILI, many of the available studies characterize liver cytotoxicity, functionality, and morphology and have limited capacity to distinguish the mechanisms behind DILI. This review aims to discuss the current understanding of mechanisms behind idiosyncratic DILI (iDILI) and the recent advancements in in vitro models, emphasizing the role of immune-mediated mechanisms using co-cultured 2D and 3D models properly incorporating non-parenchymal cells (NPC).

## 2. Mechanism of DILI in Humans

DILI is classified into two categories, dose-dependent and predictive vs. idiosyncratic and dose-independent, reflecting how drug-induced liver toxicity can provoke a wide array of multifactorial clinical presentations including cholestasis, acute liver failure, vanishing bile duct syndrome, hepatic necrosis, cirrhosis, cholestatic hepatitis, nonalcoholic fatty liver, and lactic acidosis. While intrinsic DILI drugs such as acetaminophen, amiodarone, and valproic acid elicit dose-dependent liver injury in a predictable manner, the majority of unpredictable DILI is idiosyncratic. IDILI includes immune-mediated and non-immune-mediated mechanisms. Immune-mediated mechanisms of iDILI can be phenotypically similar to idiopathic autoimmune hepatitis with histology features of hepatitis and plasma cell infiltration. A major risk factor identified for iDILI is the association with specific human leukocyte antigen (HLA) haplotypes [17]. Additional polymorphisms implicated in immune-related signaling associated with iDILI include but are not limited to a missense variant in protein tyrosine phosphatase non-receptor type 22 (PTPN22) and interleukin (IL)-10 allele [18,19].

IDILI can be further classified based on the ratio of alanine transaminase (ALT) to alkaline phosphatase (ALP), with a ratio > 5 indicating hepatocellular, < 2 representing cholestatic, and between 5–2 suggesting mixed disease [20,21]. Hepatocellular iDILI is the most prevalent form of iDILI and histologically demonstrates monocytic inflammatory infiltrates similar to viral hepatitis as part of the pathophysiology. This type of iDILI is supported by the predominance of CD8+ cytotoxic T-cell infiltration in patients, initiating or perpetuating idiosyncratic liver injury, whereas macrophage and neutrophil infiltration in predictive DILI is often considered a late-stage event as demonstrated by acetaminophen overdose [22,23,24]. While the onset of immune-mediated iDILI can be longer, the typical onset is 1–3 months after drug administration [20].

The influence of inflammatory stress may enhance the sensitivity of the dose-response curves for toxicity at recommended dosages, making it possible for features of dose-dependent predictive DILI and iDILI to overlap [15,25,26]. This complexity in disease pathogenesis and limited knowledge of the underlying mechanism is in part due to the intricate interaction between the innate and adaptive immune cells within the liver [27,28,29]. The current mechanistic understanding of iDILI includes activation of the adaptive and innate immune response in part by danger-associated molecular pattern molecules (DAMPs) and drug/metabolite exposure and collectively leading to (1) reactive metabolite accumulation and cholestasis, (2) mitochondrial dysfunction, (3) bile salt export pump (BSEP) inhibition, and (4) lysosomal impairment (Figure 1). While these events may represent the primary mechanisms associated with iDILI, it is noteworthy that other hypotheses have been proposed such as non-covalent interactions inducing immune responses, endoplasmic reticulum stress due to the binding of reactive metabolites leading to the unfolded protein response, as well as reduced synthesis of protein and the imbalance between reactive metabolites and endoplasmic reticulum chaperones [30].

### 2.1. Immune-Mediated Mechanism

Accumulating evidence indicates that the majority of iDILI cases are mediated by immune mechanisms and the interactions of these cells with parenchymal cells with few exceptions. For instance, the association with the HLA haplotype and other immune-related genes suggest that there is sufficient evidence to warrant genetic studies [17]. Though not all iDILI is linked to HLA haplotypes and potential non-immune-mediated mechanisms exist such as direct damage to hepatocytes, interference with transporters, and alterations of bile ducts which have been associated with drugs such as diclofenac that elicits DILI [31,32], studies suggest that drugs impacting hepatocytes may initiate the immune response mediated by DAMPs [33]. Briefly, the hepatic immune system involves an interplay between the innate and adaptive immunity including Kupffer cells (KCs) which act as the macrophages of the liver, dendritic cells (DCs), natural killer (NK) cells, natural killer T (NKT) cells, CD4+ helper T cells and CD8+ T cells, coordinating a rapid and robust immune response [28,29]. Both CD4+ and CD8+ T cells promote adaptive immune response with CD8+ T cells mediating the majority of injury whilst all listed immune cells are critical in host defense against pathogens and neoantigens. Early essential steps involved in this mechanism likely require the activation of antigen-presenting cells by DAMPs [11]. KCs are essential to the first line of defense via recognition of pathogens and necrotic cells. KCs release diverse signaling molecules such as IL-6, tumor necrosis factor alpha (TNFα), and chemokines, influencing cell viability, proliferation, and DME activity [34]. Additionally, the secretion of lipopolysaccharide (LPS) from the gastrointestinal tract to the portal vein activates KCs by binding to LPS-binding protein and provoking the signaling cascade to CD14 and downstream toll-like receptor 3 (TLR4), producing a pro-inflammatory response [35,36]. Simultaneously, hepatic sinusoidal endothelial cells can upregulate adhesion molecules provoked by the released cytokines, which cooperate with KCs to further release signaling molecules and factors that will maintain inflammation and promote survival [37]. The underlying role of the innate and adaptive immune response in iDILI is unclear; however, the extremely complex coordination of systems suggests that immune dysregulation influences the risk of iDILI development [38].

Historically, the withdrawal of drugs from the market due to DILI preceded the understanding of the immune-mediated mechanisms. For example, trovafloxacin was withdrawn from the market in 1999 due to acute liver failure and hepatitis (Table 1). It is now speculated that the hepatotoxicity is a result of the subsequent immune damage to hepatocytes and macrophages. Other medications such as ximelagatran, an anticoagulant which was voluntarily withdrawn from the global market by AstraZeneca in 2006, exhibit immune-mediated hepatotoxicity that may be predisposed by allelic variants in the HLA major histocompatibility complex (MHC) class II region, providing a platform to recapitulate known genomic associations [39]. Similarly, research has found that lumiracoxib, removed from the Australian market in 2007, has a strong association between the hepatotoxicity and HLA alleles within the MHC class II region, for example, DQA*0102, inferring that these HLA alleles have higher sensitivity in the identified 41 cases of lumiracoxib-induced liver injury compared to control lumiracoxib-exposed liver tissue [40,41]. These examples highlight why it is critical to include immune cells and other parenchyma cellsl/NPC in a microphysiological model to effectively investigate the interaction of the immune-mediated and non-immune-mediated mechanisms of iDILI.

### 2.2. Reactive Metabolite Accumulation & Cholestasis

The physicochemical properties of a drug can impact the cellular uptake, ADME, and, more specifically, the metabolism and conjugation by Phase I and II enzymes, overall influencing drug toxicity [54]. While many conjugated metabolites have increased water solubility for excretion, DME can also lead to the production of electrophilic chemically reactive metabolites (CRMs) and unstable conjugates, or be the target of drug toxicity. These metabolites can covalently bind to cellular proteins, form drug-protein adducts, and cause liver injury through various mechanisms [55,56]. Drugs that produce CRMs provide potential evidence for the association between reactive metabolite formation and iDILI [57]. For instance, isoflurane, although structurally similar to halothane, one of the first drugs associated with iDILI, has significantly less reactive metabolite formed and subsequently lower risk of iDILI compared to halothane [58,59]. Another prototypical DILI inducer is bromfenac, withdrawn from the US market in 1998 due to hepatotoxicity (Table 1). It is speculated that the hepatotoxicity of bromfenac is a result of a derivative of Phase II metabolism, becoming activated to acyl-coenzyme A thioesters causing enhanced reactivity, forming adducts with proteins and depleting glutathione (GSH), and ultimately leading to apoptosis [45,46].

Cholestasis includes the damage of cholangiocytes due to canalicular secretion of reactive metabolites or the imbalance of available GSH and glucuronide conjugates, reducing bile flow and impairing bile excretion [60]. Early preclinical studies leveraging rodent hepatocytes identified the link between toxic accumulation of bile acids and oxidative stress-mediated focal necrosis. Focal hepatic necrosis during obstructive cholestasis has shown evidence of extensive neutrophil-mediated oxidative stress via the formation of hypochlorus acid [61,62]. Despite several lines of evidence suggesting that reactive metabolites may play a mediator role in iDILI, no defined correlation has been established between reactive metabolites in experimental conditions and the incidence of iDILI in humans [63,64]. For instance, it has been demonstrated that the formation of reactive metabolites and metabolism-dependent covalent adducts cannot discriminate against hepatotoxic vs. non-hepatotoxic drugs in vivo following an analysis of nine hepatoxic and nine non-hepatotoxic drugs in liver microsomes [65]. Using a larger collection of drug candidates, an additional study found no clear correlation between hepatotoxicity observed in preclinical animal models and the extent of covalent binding among 100 drugs, despite drugs within that collection eliciting distinct hepatotoxicity (e.g., temafloxacin and famotidine). However, the circumstantial evidence has led some drug companies to screen drug candidates for covalent binding [66].

### 2.3. Mitochondrial Dysfunction

Mitochondrial dysfunction is an encompassing term for alterations in metabolic pathways and damage to mitochondrial components, consequentially causing oxidative stress, disruption in energy homeostasis and the mitochondrial membrane permeabilization, accumulation of triglycerides, inhibition of fatty acid ß-oxidation, and, ultimately, cell death. Apoptosis-inducing drugs alter mitochondrial energy homeostasis, via damage to mitochondrial subunits, and play a vital role in the pathogenesis of iDILI [67,68]. Mitochondrial damage can lead to hepatic apoptosis and necrosis, signaling through the c-Jun N-terminal kinase (JNK) pathway, and subsequent cell death [69]. Additionally, the mitochondrial electron transport chain can be impaired, leading to iDILI [70]. Drugs, such as valproic acid, biguanides, and diclofenac, impair mitochondrial respiration and disrupt the integrity of the mitochondrial membrane, suggesting this mechanism is implicated with their effect on liver injury [71,72]. A prototypical DILI inducer withdrawn from the US market due to hepatotoxicity before being re-approved is Tolcapone. This drug is a catechol-O-methyltransferase (COMT) inhibitor that selectively and reversibly inhibits COMT, reducing the catabolism of levodopa to 3-O-methyldopa and increasing overall dopamine available in the brain [73]. Tolcapone is linked to mitochondrial impairment associated with iDILI and this toxicity differs from second-generation COMP inhibitors, possibly because tolcapone is oxidized into an amine or acetylamine metabolite and forms reactive species [48,49] (Table 1). While it took decades to understand the role mitochondrial damage played in iDILI, predictions based on an analysis of 124 compounds has led to the strong association between the loss of the integrity of the mitochondria and the risk of iDILI [74]. Thus, it is no surprise that many of the cellular mechanisms potentially involved in iDILI are overlapping in nature, complicating the investigation of distinct mechanisms.

### 2.4. BSEP Inhibition

In addition to impairment of the mitochondria, known compounds that cause liver injury lead to physiological alterations in hepatic bile acid homeostasis. While there are various transporter proteins that regulate the uptake and efflux of drugs/metabolites, the inhibition of the ATP-dependent BSEP, also called ABCB11, has been most notably implicated in iDILI [44]. Specifically, the BSEP transporter is responsible for the biliary excretion of bile acids and, thus, impaired function causes the accumulation of cytotoxic hepatic bile acids, oxidative stress, and cell death through apoptosis and necrosis pathways [75]. Using BSEP-inverted vesicles, Morgan et al. demonstrated the potential risk of iDILI from drugs/metabolites, such as troglitazone, that inhibit BSEP [44]. However, a recent report suggests that in vitro measurements of BSEP activity are not useful in the prediction of iDILI, because the majority of the evidence on the topic was obtained in vitro at higher-than-clinically-relevant concentrations [76]. Further complicating the in vitro study of BSEP inhibition as a mechanism for iDILI, metabolites such as the sulfate conjugate of troglitazone may have greater inhibition of BSEP compared to the parent compound [77]. Sitaxentan, an endothelin receptor antagonist intended to treat pulmonary arterial hypertension, is another example of an iDILI-causing drug which was removed from the US market in 2010 that is associated with BSEP inhibition [53] (Table 1). Repression of BSEP expression may play an important role in drug-induced cholestatic liver injury beyond direct inhibition [78,79,80]. However, it is noteworthy that mechanisms are multifaceted. For instance, troglitazone, a drug that was recalled from the US market due to DILI and now used as a prototypical iDILI inducer, has been proposed to cause hepatotoxicity via (1) the formation of reactive quinones and quinone methides leading to subsequent GSH depletion [42], (2) the depletion of ATP and cytochrome c release causing apoptosis [43], and (3) BSEP inhibition by the parent compound and troglitazone sulphate metabolite leading to cholestasis [81] (Table 1).

Recent studies have also connected the inhibition of other hepatobiliary transporters such as multidrug resistance protein (MRP)2/3/4 (ABCC2, ABCC3, and ABCC4), which are responsible for excreting bilirubin, drug metabolites, and conjugated organic anions [82]. Clearly, additional factors should be considered beyond transporter inhibition. For instance, drugs that elicit severe DILI may have multiple mechanisms including BSEP inhibition, mitochondrial impairment, and both immune-mediated and non-immune-mediated mechanisms [70,83].

### 2.5. Lysosomal Impairment

Drugs that provoke mixed forms of fat accumulation in the liver can cause macrovascular and microvacuolar steatosis in hepatocytes due to the inhibition of mitochondrial beta-oxidation and oxidative phosphorylation [84]. Some drugs that are responsible for hepatosteatosis can cause latent forms of nonalcoholic steatohepatitis (NASH) or hepatotoxicity. Typically, these cationic amphiphilic compounds with an amine group can become protonated and subsequently uncharged, as the drug traverses into a lysosome/outer membrane of the mitochondria and into the acidic interstitial membrane space. Ultimately, such drugs can get trapped in the lysosome because they cannot cross back following protonation. This change of intracellular distribution causes drugs and their noncovalent complexes with phospholipids to accumulate in the lysosome, known as phospholipidosis [85,86]. These cationic amphiphilic compounds can also impact the mitochondrial function and electron transport chain by crossing into the outer membrane of the mitochondria, becoming protonated, and passing into the inner mitochondrial matrix where compound accumulation occurs. For instance, the iDILI of amiodarone is associated with its high propensity to accumulate in mitochondria and inhibit phospholipase activity [87,88].

### 2.6. Regulatory Considerations for DILI per Guidance for Industry

It is noteworthy that in vivo animal models have been used to investigate iDILI and have long been considered the gold standard in drug development [12,89,90]. Although animal models continue to be immensely useful in understanding the interaction between immune cells and parenchymal cells, there is little consistency in the drugs that are known to cause iDILI in humans and the drugs that elicit liver injury in animals, even at concentrations markedly higher than those used in humans. These contradictory results highlight the need for alternative models that allow further investigation, adjustable complexity, and complementary assessment to in vivo models. Moreover, regulatory agencies such as the FDA have outlined a need to target investment and research into fostering the development of potential applications of alternative systems such as microphysiological systems, in vitro and in vivo modeling, and alternative methods in toxicology, to inform regulatory decision-making. Additionally, the FDA Omnibus Reform Act of 2022 (FDORA) encourages drug developers to use a number of alternatives to animal testing, including cell-based assays and computational models, to investigate the safety and efficacy of an investigational new drug when suitable.

Consistent with the attention on alternative methods for understanding toxicology, it is noteworthy that the FDA Guidance for Industry, titled “Drug-Induced Liver Injury: Premarketing Clinical Evaluation”, describes that hepatocellular injury is necessary but not sufficient to identify DILI in drug development programs. Further, elevation in the serum transaminase concentration may not only be an indicator of severe DILI in extremely high levels and instead, may be an indicator of altered liver function. Thus, Hy’s law is used to predict severe DILI by measuring (1) hepatocellular injury by three-fold greater elevation above the upper limit normal (ULN) in alanine aminotransferase (ALT) or aspartate aminotransferase (AST), (2) the elevation of total bilirubin levels two-fold above the ULN without initial indication of cholestasis, and (3) no other reason for the elevation in transaminases or bilirubin levels. This method of detecting potential hepatotoxic drugs has proven useful previously and is part of the safety evaluation of many marketed drugs. For example, nefazodone was withdrawn from the US market in 2004 due to acute liver failure (Table 1). Further research from histology preformed on livers exhibiting nefazodone-mediated hepatic injury suggest bile-duct proliferation with cholestasis and increased ALT, AST, and total bilirubin > 10 × upper limit normal [91]. It has been speculated that a potential mechanism of nefazodone-induced hepatotoxicity is linked to the inhibition of BSEP [50], as well as impairment to mitochondrial respiration via oxidative phosphorylation of complexes I and IV and reactive intermediaries [51,52]. While AST and ALT levels can be adequately measured in animal models, adequate measurement of bilirubin levels presents a particular challenge. However, in vitro models offer the potential to overcome this obstacle through the testing and predictive modeling of prototypical hepatotoxic compounds such as nefazodone using insight from their potential mechanisms to recapitulate physiologically relevant DILI.

While Hy’s law is used in the FDA Guidance to Industry to assess the potential for a drug to cause severe liver injury, the European Association for the Study of the Liver (EASL) has published clinical practice guidelines which characterize ‘hepatocellular’ liver injury as a five-fold or higher increase in ALT alone or when the ratio of serum activity of ALT to alkaline phosphatase (ALP) is 5 or more, ‘cholestatic’ liver injury as a two-fold or higher increase in ALP alone or when the ratio of serum activity of ALT to ALP is 2 or less, and ‘mixed’ when the ratio of the serum activity of ALT to ALP is between 2 and 5 [92]. Thus, using additional serum aminotransferases to capture DILI may enable further biomarkers to validate a DILI method during drug screening. 

## 3. Overview of Hepatocyte–NPC Co-Culture Models

The complexity of the liver has been challenging to recapitulate in a single in vitro model capable of exhibiting all the critical metabolic features. With the desire to control complexity and gain insight into the pathogenesis of iDILI, some studies have utilized reactive iDILI-associated media and cytokine conditions to culture hepatocytes instead of co-culturing immune cells with hepatocytes [93,94]. For instance, Melino et al. observed phenotypic and functional alterations in hepatocytes following the use of media conditioned by THP-1 macrophages [93]. Conversely, models that study the impact of drug-exposed hepatocyte media on immune cells have also been used [37,95]. Adding hepatotoxicant-conditioned media to KC reduced the secretion of anti-inflammatory cytokines IL-6 and IL-10 while increasing the production of pro-inflammatory IL-1β [96]. However, insufficient crosstalk among the immune cells and hepatocytes and lack of direct cell interaction have limited the use of these models. Overall, studies using conditioned media have suggested that the hepatotoxicant or DAMPs released by hepatocytes can trigger an immune response, yet the presence of both immune cells and hepatocytes are necessary to illuminate the complex crosstalk, cell interaction, and signaling that occurs during iDILI.

Thus, to detect iDILI with maximal desired features, hepatocytes (the major parenchymal cell of the liver) and NPCs are often co-cultured to improve the sensitivity and specificity of the detection of iDILI-inducing compounds. With complexity ranging from simple monocultures to organoids, these models intend to recapitulate functional aspects of the liver. Several in vitro human-based models are used to predict iDILI and the various phenotypes with an assortment of cell sources and endpoints. Models can use various cell types, such as liver slices and isolated perfused livers, human hepatoma cell lines (e.g., HepG2, Huh7, and HepaRG), primary hepatocytes, and stem-cell-derived hepatocytes. Among others, liver slices and derived liver tissue contain both NPC and parenchymal cells that achieve the physiologically relevant architecture of the liver and retain the metabolic function [97,98]. However, within hours, the metabolic capacity and function decrease, and the hepatocytes start to die, thus limiting the application.

Alternatively, cell-based experiments control complexity and are adaptable to the high throughput screening of hepatotoxic compounds. Fresh and frozen primary hepatocytes are essential for establishing a physiologically relevant liver model, largely due to their metabolic competence and retained hepatic functionality. However, the availability of primary human hepatocytes can be limited and de-differentiation can occur rapidly in two-dimensional (2D) cultures [99].

To overcome availability, cost, and other limitations, hepatoma cell lines are sometimes used due to the ease of culture techniques and their unlimited cell numbers. The most commonly used immortalized liver cells have varying degrees of metabolic capability (HepG2, HepaRG, and Huh7). Compared with HPH, immortalized human liver cell lines, except HepaRG (a surrogate of HPH), represent different stages of cellular phenotype and metabolic activity, and express dissimilar DMEs and transporters, drastically reducing their value as a DILI model [100,101]. In contrast, properly cultured HPH often recapitulate liver physiology in a more representative manner than hepatoma cell lines [102]. Additionally, the ability to cryopreserve HPH with preserved metabolic competence and drug sensitivity has afforded physiologically relevant reproducibility in the same donor and reduced genetic variation among experiments [103,104]. Importantly, a major challenge to the application of fresh human hepatocytes is that they are not phenotypically or metabolically identical to in vivo conditions, as a traditional monolayer of 2D hepatocyte culture quickly loses metabolic capability. The integration of NPCs in hepatocyte cultures in various forms has stabilized the function of HPH for long-term culture. With the advancement in culture techniques, cryopreservation protocols, and the overall understanding of the complex liver system, new platforms are being used to investigate and improve hepatocytes in combination culture with other NPCs (Figure 2) [105].

### 3.1. Two-Dimensional Co-Cultures

An abundance of in vitro co-culture models utilize KCs, which are the primary resident hepatic macrophages [106]. Other NPCs such as hepatic stellate cells (HSC) and liver sinusoidal endothelial cells (LSEC) are also frequently co-cultured with hepatocytes in combination with KCs. Interestingly, differences in HPH:KC ratios dependent on the inflammatory state further complicate model design [107,108]. For instance, to recapitulate inflammatory states, the ratio of HPH to KCs at 2.5:1 has been used [108]. Other studies have used a HPH:KC ratio of 10:1 to mimic that KCs make up 10% of the liver under normal condition [109,110]. Differences in the ratio of HPH:KCs are important factors for the consideration of supporting an inflammatory condition in vitro that matches the in vivo physiology. Due to the high cost and lack of availability for certain NPCs, studies have alternatively used hepatoma cell lines and THP-1 monocyte/macrophages for co-cultures [111,112].

Fueled by the improvements in cryopreservation, culturing, and isolation of HPH, mounting efforts have optimized the longevity of the monolayer 2D cultures, including the development of the extracellular matrix allowing for a sandwich format with collagen and a Matrigel overlay [113]. HPH is now co-cultured in 2D formats, including mixed cultures [114], micropatterned [36,115,116] and transwell formats [117,118], and chemically modified culture media [119], attempting to retain their hepatic function and morphology for longer than conventional culture conditions.

#### 3.1.1. Mixed Co-Cultures

Co-cultures with hepatic cells and immune cells have offered advantages in deciphering the complex and dynamic cellular interactions that are not possible using a hepatocyte monolayer. Thus, indirect hepatocellular toxicity provoked by immune-mediated mechanisms is often disregarded in monoculture drug screening [120]. The development of optimized co-cultures that incorporate NPCs into hepatic cell culture potentially enables the study of iDILI and the role of the immune response [121]. Several studies have focused on optimizing the co-culture conditions, taking into consideration the NPC fraction [117,122,123] for KCs [116,124], HSC [124,125], and LSEC [126,127].

Metabolically competent in vitro mixed co-cultures can evaluate and predict potential compounds that are associated with iDILI. For instance, Rose et al. assessed CYP3A activity and cytokine profiles (IL-6 and TNFα) following trovafloxacin, an antibiotic associated with immune-mediated iDILI, and acetaminophen, an analgesic that converts to reactive metabolites associated with dose-dependent DILI, in a donor-matched HPH-and-KC mixed co-culture [114]. The authors established culture conditions that retained longevity and hepatocyte function and found that trovafloxacin led to a concentration-dependent change in the IL-6/TNFα ratios and shifted the CYP3A inhibition mediated by IL-6. Additionally, in the co-culture that was stimulated by LPS and treated with acetaminophen, there was a concentration-dependent reduction in IL-6 concentrations that coincided with a significant increase in CYP3A activity without altering TNFα levels [114]. However, this study does not include a direct cytotoxicity assessment, and the impact of the reduction of IL-6 and CYP3A activity on cell viability cannot be concluded. 

However, a major challenge with mixed co-cultures is that the arrangement of NPC and hepatocytes and the ability to individually assess each cell type cannot be controlled. Additionally, a single NPC type does not account for the complex cohesion and cell–cell interaction that occurs in vivo. Interestingly, a collagen sandwich culture that incorporates rat stellate cells on top of the collagen gel, then rat hepatocytes, followed by LSECs and KCs cultured in a fibronectin-coated transwell was generated to evaluate inflammatory response and NPC-hepatocyte interactions [128]. While no drugs were tested with this model, the authors found that co-cultures stimulated with LPS had triggered a TNFα response, and inhibited CYP activity and albumin production, suggesting a feedback mechanism that decreased metabolic activity and hepatic function. While this model has been used to advance findings on liver physiology, pathophysiology, and the role of the adaptive immunity on hepatic function, more sophisticated and complex models have become a major focus and critical to predicting iDILI.

#### 3.1.2. Micropatterned Culture

In an effort to control the cellular microenvironment, enhance the function of parenchymal cells and NPCs, and improve drug sensitivity, micropattern co-cultures have been generated and extensively evaluated. Although the exact mechanism underlying their complex interaction remains unclear, these co-cultures have provided meaningful biological evidence for xenobiotic metabolism, drug toxicity such as DILI, lipid metabolism, and stress responses, to name a few. Physiologically relevant models exploit the fact that cells are not randomly distributed and that they require heterotypic cell–cell interactions. The first micropattern co-culture seeded rat hepatocytes on a collagen-coated disc and surrounded the hepatocytes with 3T3-J2 murine fibroblasts [129,130]. This first-of-their-kind co-cultures allowed the ratio and cell number to be consistent across different patterned configurations. Collectively, this innovative study pioneered key takeaways such as (1) circular platforms provide better cell retention and fidelity over longer periods of time than rectangular platforms, (2) liver-specific function cannot be recovered from homotypic interactions alone, (3) hepatocyte metabolic function could be improved by reducing the diameter of the collagen-coated disc, and (4) the heterotopic interaction between fibroblasts and hepatocytes were required to retain the hepatocyte phenotype, and conditioned media with fibroblasts was insufficient [105].

Micropatterned co-cultures display phenotypic stability for weeks with sustained liver-specific function. Further studies have evaluated albumin, urea, ATP, and glutathione (GSH) as hepatotoxicity endpoints. For instance, Khetani et al. screened 45 drugs from a database with 35 drugs known to cause DILI (29 are known to cause iDILI specifically) and 10 negative controls using human hepatocytes and rat hepatocytes co-cultured with stromal fibroblasts [131]. It reveals that albumin was the most sensitive marker for hepatotoxicity, followed by urea secretion and ATP, and GSH was the least sensitive. Human hepatocytes had a higher sensitivity to drugs (65.7%) shown to cause iDILI compared to rat hepatocytes (48.6%).

Additionally, micropatterned co-cultures show significantly higher predictive capability than a conventional HPH monolayer. For instance, in a study that used an HPH micropatterned co-culture compared to an HPH conventional co-culture, the micropattern co-culture correctly identified the toxicity of fialuridine, an agent intended for hepatitis B infection but caused liver failure and fatal lactic acidosis of five patients within the clinical trial [132]. Further studies have elevated micropattern co-cultures to utilize NPCs and hepatocytes and maintain cell polarity and superior metabolic function compared to HPH monolayers [116]. In this case, HPH were pre-established in the micropattern co-culture for 5–7 days, and the KCs were seeded and stimulated with bacterial-derived LPS and endotoxin, to exhibit a pro-inflammatory and down-regulated metabolic phenotype in HPH [36]. Moreover, micropatterned co-cultures with primary hepatic stellate cells and HPH have been used to determine hepatic function following treatment with NASH modulation compounds [133].

#### 3.1.3. Transwell

Studies have also utilized hepatocytes or hepatoma cells separated by a porous membrane to investigate drug toxicity. Granitzny et al. developed a transwell iDILI model combining HepG2 and monocytic/macrophage-like THP-1 cells to mimic immune infiltration [111]. Four drug pairs within the same drug class were tested where one drug in the pair is associated with iDILI and one is not (e.g., troglitazone–rosiglitzaone, trovafloxacin–levofloxacin, diclofenac–acetylsalicylic-acid, and ketoconazole–fluconazole). Transwell co-cultures were exposed to LPS, TNF, or a vehicle control, and the HPH viability was tested following treatment. Results from this study suggest that all tested drugs associated with iDILI led to synergistic cytotoxicity upon exposure to an inflammatory microenvironment, supporting an involvement of inflammatory stress in the development of iDILI [111].

Two other studies investigated if ketoconazole and troglitazone provoked iDILI in two transwell co-culture models, one with Huh7 and THP-1 [134] and the other with HepG2 and THP-1 cells [112]. The Huh7 and THP-1 transwell co-culture detected an increased expression of pro-inflammatory cytokines/chemokines (e.g., CXCL2, CXCL10, IL-6, and IL-1β) and increased cytotoxicity following troglitazone treatment compared to the monolayer co-culture [134]. While the mechanism behind troglitazone-mediated iDILI has not been fully elucidated, pro-inflammatory cytokines have been implicated. However, these authors only observed a modest increase in these pro-inflammatory cytokines after troglitazone treatment consistent with results in prior studies [135,136]. Nonetheless, the co-culture demonstrated significantly higher chemokines, CXCL2 and CXCL10, indicating a potential inflammatory response compared to the conventional monoculture. Additionally, Wewering et al. used HepG2 and differentiated THP-1 cells to evaluate ketoconazole-induced hepatotoxicity and the subsequent secretion of 36 different pro-inflammatory cytokines (e.g., CXCL8, TNFα, and CCL3) in the co-culture compared to the monoculture of each cell type [112]. The co-culture presented increased levels of macrophage inhibitory factors (MIF) CCL3, CD54, GM-CSF, CXCL8, TNFα, IL-1rα, and serpin E1, while the monoculture of THP-1 cells detected MIF, CCL3 and IL-rα only. Similarly, it was shown that the co-culture treated with ketoconazole provoked the secretion and potential activation of the CXCL8 pathway, a pathway critical in the regulation of inflammatory genes, while the monoculture did not induce the mRNA expression for CXCL8, TNFα, CCL5, and CD54. Continuing innovation into the modeling and prediction of iDILI have merged transwells with 3D scaffolds to more closely resemble in vivo settings compared to 2D transwells.

### 3.2. Three-Dimensional Co-Cultures

A main limitation of 2D conventional monolayer models, even with the presence of an extracellular matrix, is the morphology and functional decline of HPH, making chronic drug evaluation difficult. Additionally, none of the hepatoma cell lines under 2D conformation have sufficient hepatocyte function, phenotype, and metabolic capacity to fully predict compounds that elicit hepatotoxicity. In contrast, three-dimensional (3D) spheroids and organoids retain their metabolic activity, physiologically relevant expression of DMEs and transporters, biotransformation capacity, and sensitivity to drugs in comparison to 2D cultures [137,138]. Recent advances in 3D models have led to a shift from 2D conventional cultures to complex 3D approaches that employ multicellular microphysiological devices to recapitulate characteristics of in vivo human tissues [139,140]. This shift is largely driven by the ability to adapt 3D models to high-throughput screenings with a variety of complexity and scalability. These models include but are not limited to spheroid and organoid co-cultures, bioprinted systems, perfusion-based models such as bioreactors, and microfluidic co-cultures.

#### 3.2.1. Spheroid NPC Co-Cultures

Liver spheroids are reported to retain 3D architecture, cell–cell interaction, and viability for extended periods of time (up to 5 weeks) [137,141]. Likewise, transcriptomic and proteomic analyses have identified that liver spheroids have similar characteristics to in vivo liver conditions compared to the 2D format, providing a more complex iDILI prediction model [142,143]. Briefly, it was observed that 97.5% of mRNA transcripts and 92.7% of proteins were stable in spheroids containing HPH and NPCs over 35 days detected via microarrays and proteomics, particularly compared to hepatoma cell lines and liver slices [143,144]. Notably, 3D spheroid cultures of hepatocytes alone remain viable long-term and maintain relatively stable transcriptomes/proteomes; however, there is a considerable reduction in the expression of DME and drug transporters. For instance, CYP2C8 and CYP2E1 rapidly decreased expression, with a reduction in expression of 10-fold less by day 35, while MRP2, P-gp, and vimentin increased expression from day 1–7 and then plateau and remained stable [143]. On the other hand, co-cultured spheroids with hepatocyte-like cells and LSECs have higher CYP activity and urea secretion compared to spheroids with hepatic cells alone [145]. Accumulating evidence indicates that the addition of endothelial cells can improve the functional maturity of hepatocytes and promote polarization and the 3D architectural arrangement [146,147].

NPCs co-cultured with parenchymal cells in spheroids are used to recapitulate in vivo conditions more closely than HPH alone. KCs are a key component to the innate immunity; however, the role of KCs and other NPCs in iDILI is poorly defined.

Spheroids can be generated in a variety of techniques, with and without scaffolds, by hanging-drop methods, hydrogels, and nanoimprinted structures [137,148,149]. Scaffold-dependent methods rely on synthetic or biological scaffolds/matrix to grow spheroids, aiding in the architectural arrangement of the cells. Biological scaffolds assist in the recapitulation of the in vivo cell environment by providing an extracellular matrix and interaction between cells, mimicking cell signaling, behavior, and survival [150]. However, scaffolds can also impact the cellular phenotype and enable the binding of compounds to the scaffold; particularly, synthetic scaffolds may generate toxicity associated with the material. Synthetic scaffolds also differ in the degree of synthetic material used such as fully synthetic constructs [151], semi-synthetic [152], and decellularized liver tissue [153]. On the contrary, scaffold-free models permit spheroid formation entirely based on self-assembly, forming cell aggregates, as in the case of ultra-low adhesion plates [154]. Bell et al. previously showed that ultra-low attachment plates preserved the function of cryopreserved HPH spheroids and enabled the study of hepatitis, steatosis, cholestasis, and iDILI, constituting a promising iDILI model that is adequate for long-term dosing and extended culture [137]. Additionally, HPH spheroids with the incorporation of NPCs such as KCs, stellate cells, and LSECs have been used to improve the detection of iDILI, increasing predictive values and sensitivity for drugs that elicit hepatotoxicity [137,149,155,156]. Proctor et al. demonstrated that 3D human liver multicellular spheroids, composed of HPH, KCs, and LSECs, have enhanced sensitivity to identifying hepatotoxic drugs compared to a 2D HPH monoculture, after screening a panel of 110 marketed drugs (63% were associated with DILI, and 37% were not associated with DILI) [155]. Eight concentrations of compounds were treated for 14 days or 2 days to mimic chronic and acute toxicity, demonstrating that spheroids outperformed HPH monocultures in differentiating hepatotoxicants from different classes and demonstrating sufficient capacity to measure liver injury markers such as HMGB1, miR-122, and α-GST [155].

Additionally, following the treatment of 100 drugs that elicit iDILI, 3D HPH spheroids with repeat dosing were more sensitive to detecting compounds that have no iDILI concern, enzyme elevation, and low iDILI concern than 2D monolayer cultures from the same donor lot, while the 3D spheroids had increased false positives for drugs that elicit a high iDILI concern and severe iDILI compared to the 2D culture [157].

Using co-culture spheroids with HPH and KCs to evaluate 14 DILI-positive compounds, Li et al. observed that IL-6 secretion was recapitulated in the co-culture spheroids stimulated by LPS, depicting an inflammatory response [157]. Following a 48-hour treatment of trovafloxacin as a prototypical iDILI inducer, the co-culture spheroids showed significantly augmented cytotoxicity and decreased IL-6 secretion compared to the HPH spheroids which declined after a 5-day treatment. Moreover, following acetaminophen treatment, co-culture spheroids were more sensitive to acetaminophen-induced toxicity compared to the HPH spheroids, depicted in a shift in the IC50 from 2247 µM to 3280 µM, whereas co-culture spheroids were desensitized by LPS stimulation (IC50 2462 µM with no LPS to 4642 µM with LPS) [157]. This altered sensitivity suggests KC activation may play a protective role in acetaminophen toxicity. Collectively, this evidence suggests that hepatocyte–NPC co-culture spheroids are an attractive platform to model iDILI, although the drugs/compounds screened thus far are limited. Hence, further investigation is warranted to evaluate the predictive capability of hepatocyte–NPC co-culture spheroids for iDILI that can be applied to diverse DILI-associated compounds in different pharmacological classes.

#### 3.2.2. iPSC-Derived Liver Organoids

Recent advancement in stem cell differentiation has shown the potential to utilize induced pluripotent stem cell (iPSC)-derived cells to recapitulate the liver microenvironment and model liver development and pathophysiology. The self-assembled 3D liver organoids reprogrammed from iPSC or adult stem cells contain functional hepatocyte-like cells and NPC cells in a proportion that mimic their in vivo counterparts. With the recent advances in iPSC reprogramming, more mature functional liver cells, recapitulating the cellular heterogeneity and important cell–cell and cell–extracellular-matrix interactions, have been generated using different differentiation protocols [158]. Transcriptomic analyses revealed that the liver organoids express a gene profile mimicking that of in vivo livers, including genes encoding cytochrome P450 enzymes, albumin, CK18, alpha-1-antitrypson, and glycoprotein critical for liver function [159,160].

In addition to studying liver development and disease modeling, iPSC-derived human liver organoids have proven to be an attractive model for DILI investigation. Holmgren et al. found that iPSC-derived hepatocytes are sensitive to the hepatotoxicity of amiodarone, aflatoxin B1, and troglitazone [161]. In varying levels of complexity, iPSC-derived liver organoids offer an individualized approach to testing iDILI in the preclinical phases of drug development. For instance, Shinozawa et al. tested 238 marketed drugs in four concentrations on liver organoids generated from 10 different iPSCs that demonstrated high predictability (sensitivity: 88.7% and specificity: 88.9%) for cholestasis, bile salt accumulation, and/or mitochondrial dysfunction, thus addressing some of the mechanisms behind DILI [162]. Additionally, Koido et al. explored a ‘polygenicity-in-a-dish’ strategy using genetic-, cellular-, organoid- and human-scale evidence to predict potential iDILI susceptibility in humans and demonstrated that data from liver organoids from different donors can be informative for designing safer and more efficient clinical studies [163]. Most recently, Zhang et al. developed human-liver organoids from three separate iPSC lines over a 20-day reprogramming period to exhibit markers for hepatocytes, stellate-like cells, and KCs [164]. Using this model, albumin production, ALT and AST release, as well as viability were assessed following exposure to tenofovir and tenofovir in combination with inarigivir. The combination of tenofovir and inarigivir led to increases in ALT and AST and diminished albumin production, while no effect was observed with single agents. Although the liver organoid remains an imperfect model with limitations in cell maturation, cell-type representation, and high cost, technologies that advance the utility of iPSC-derived liver organoids present an exciting strategy for drug development while minimizing the potential DILI. Improved reprogramming approaches that overcome these limitations are expected to advance this exciting model to the next level.

#### 3.2.3. Bioprinting

Bioprinting of 3D biological tissue structures cover a variety of methodologies for depositing cells and extracellular matrices such as droplet, extrusion, and light-assisted bioprinting. Droplet-based bioprinting dispenses a precise solution of cells in small droplets, with resolution capabilities down to 2 µm (hydrogel) and 50 µm for cells [165,166]. Mastsusaki et al. utilized droplet-based bioprinting to produce a co-culture model including HepG2 and LSECs where increased albumin secretion and CYP3A4 activity was observed compared to the HepG2 monolayer [167]. Nguyen et al. used bioprinting to develop a cryopreserved HPH, stellates, and LSECs 3D tissue to evaluate the tissue-level iDILI and dose response to trovafloxacin compared to non-hepatotoxicant levofloxacin [168]. At clinically relevant concentrations, trovafloxacin exposure led to significant cytotoxicity, observed without the stimulation of LPS or macrophages. 

Extrusion-based bioprinting deposits continuous filaments and is one of the most common techniques due to its versatility and compatibility with printing scaffold-free properties [169,170]. While droplet-based bioprinting provides superior resolution, extrusion creates shear stress and is limited to printing >100 µm reducing the cytocompatibility. Remarkably, Lee et al. used extrusion-based printing to incorporate vascularization with murine primary hepatocytes, endothelial cells, and a human lung fibroblast to promote heterotypic interaction [171]. This 3D hepatic structure with NPCs demonstrated improved albumin secretion and urea synthesis over the course of 10 days with 20-fold higher values compared to a hepatocyte monoculture. Lastly, light-assisted bioprinting prints with high resolution, increased speed, and low shear stress, making it an expensive but promising platform for 3D bioprinting. Due to the high resolution, this bioprinting modality allows microscale hexagonal architecture to be generated with enhanced molecular markers compared to a monolayer [172].

Three-dimensional (3D) bioprinting is rapidly emerging as an approach for testing drug toxicity and resembles the complexity of in vivo models for preclinical assessments. However, these models are less scalable for high-throughput screening due to their difficult handling, complexity, and biosensors. At present, 3D printing is still in its infancy and demonstrates a proof-of-concept model that enables complex and dynamic cell interactions to occur and maintains metabolic function and vasculature.

#### 3.2.4. Perfusion-Based Co-Culture Models

Static co-culture models are limited in their ability to mimic dynamic cell interactions required to truly recapitulate the in vivo liver environment. Perfusion-based in vitro co-culture models have offered several advantages over static cultures, such as: (1) they enable vascular perfusion, concentration gradients, sheer stresses, and tension forces on cellular membranes, (2) they permit interaction of different cell types, and transport metabolites, signaling molecules, and chemokines/cytokines under flow, (3) they allow pharmacokinetic studies and drug-induced toxicities to be assessed, (4) and, depending on the flow system, they allow the study of permeability and barrier function with air–liquid interfaces, suggesting broader applications resembling in vivo conditions. These perfusion-based models are often automated, as in the case of the automated stirred-tank bioreactors, which can promote spheroid formation, enabling extended culture duration and repeat chronic dosing made possible by recirculation and feed modes [173,174].

While perfusion-based models are beneficial in developing a model that recapitulates a chemical gradient that is physiologically relevant, there are challenges associated with these systems that warrant consideration. For instance, the material used for generating the model, for example, a hydrophobic material such as polydimethylsiloxane (PDMS), can lead to non-specific binding of the drug to the tubing and device material. Partitioning of molecules into PDMS can reduce the concentration of drugs and potentially change experimental results; thus, this material can be a major disadvantage to these systems. Additionally, bubble formation and accumulation can lead to disruption of flow as microfluidic systems often rely on small volumes and narrow channels [175]. In addition, changes in temperature, adaptors, valves, model materials, and channel geometry can lead to cell damage and death. Moreover, the cost of perfusion-based models is usually high.

##### Bioreactors

Bioreactors, in this context, facilitate remote monitoring of cultures by controlling conditions, such as media flow, temperature, pH, glucose, lactate production, and gas tension, and stabilize the microenvironment for liver cell cultures [173,176]. Through monitoring the oxygen concentration, data can be generated about changes in metabolic activity, to deduce cell viability [177,178]. These systems function under linear or circular perfusion, allowing the continuous addition, mixing, and removal of nutrients to ensure a concentration nutrient gradient occurs to maximize hepatic metabolic function [179]. Using HPH spheroids, repeat dosing has been tested using an automated perfusion bioreactor for 3–4 weeks [173]. The spheroids in this model maintained their metabolic capacity, as assessed by the induction of Phase I and II enzyme expression and activity, and the expression of albumin, HNF-4α, and CK-18, and preserved the rate of albumin and urea synthesis. 

A common bioreactor used for recapitulating 3D perfused liver systems is the hollow-fiber bioreactor [180]. While typically used with a single hepatic cell line, the bioreactor allows parenchymal cells to be cultured surrounding extra-capillary space, providing a pseudo-vascularized model with mimicked capillary blood flow exchange between tissue. Developed by Gerlach et al., the original hollow-fiber bioreactors required 10^10^ cells, incompatible for the application of multi-drug toxicity screening [181,182]. Subsequently, miniaturized cell compartments were designed to accommodate smaller volumes of media and cells (10^7^–10^8^ cells) without compromising the metabolic activity and hepatic function [183].

In perfused bioreactors that co-culture HPH with NPC under microfluidic conditions, HPH maintain urea section and albumin production, as well as BSEP and MRP2 expression and activity [184,185,186]. The expression of CYP1A2, MRP1, and UGT1A5 remained identical to freshly isolated hepatocytes up to 13 days [187]. Using this perfused bioreactor, co-cultures of HPH and NPCs were treated with acetaminophen (0.5, 3, and 10 mM) daily for 20 days under a constant flow rate (10 μL/h) mimicking the metabolism rate and turnover, confirmed by the detection of the formation of acetaminophen glucuronide. Acetaminophen treatment led to the dose-dependent depletion of GSH and ATP at all concentrations within the HPH compartment and significantly more potently in the NPC vascular compartment. Additionally, treatment with JNJ-1, a tyrosine kinase inhibitor that targets the CSF1/CSFR pathway, depleted KC viability and subsequent cytokines [187].

In another example, 12 individual bioreactors containing a porous scaffold with HPH and NPC co-culture spheroids perfused with integrated pumps were used to assess the drug metabolism of trovafloxacin and tamoxifen following LPS stimulation [188,189]. Using this model, Rubiano et al. found that (1) lactase dehydrogenase release was increased and CYP3A4 activity was decreased following trovafloxacin treatment, (2) hepatocytes retained their metabolic function, CYP activity, and albumin secretion more than the sandwich culture, spheroids, and 2D format, at 18 days culture, and (3) the model showed the ability to quantify troglitazone metabolites, diclofenac clearance, and the intracellular accumulation of chloroquine.

##### Microfluidic Systems (Liver-on-a-Chip)

Perfusion-based systems such as bioreactors and microfluidic devices tend to overlap. For instance, hepatic cells can be cultured on a microfluidic “organ-on-a-chip” which incorporates controlled media, pumps, and nutrient supply, mimicking shear force. These microfluidic devices typically influence 3D architecture, providing layers of NPCs adjacent to hepatocytes, separated by an extracellular matrix [190]. Hepatocyte molecular phenotype and functional capacity differ in correspondence to the oxygen gradient, with the oxygen-rich periportal zone enabling enhanced urea synthesis, beta-oxidation, and gluconeogenesis, while the inverse is seen with glycolysis, bile acid synthesis, and CYP metabolism in the oxygen-poor zones [191]. Notably, perfusion modeling in these chip models is reported to allow recapitulation of the oxygen zonation and glucose concentrations, maintain expression of DMEs, and to influence and predict drug-induced hepatotoxicity [192].

Importantly, the cell-to-media ratio within these devices avoids the dilution of signaling molecules, cytokines, metabolites, and nutrients, without compromising the cell viability [193]. Haque et al. has evaluated the functional importance when accumulated cells produced elevated concentrations of cytokines and albumin secretion, CYP activity and expression, MRP-2 expression, and bile canaliculi formation, resembling in vivo levels [194]. Findings from this study suggest that the cells cultured in the microchambers synthesized secreting proteins, maintained metabolic activity via CYP1A1 and CYP3A4 induction and expression, and demonstrated hepatic functional biomarkers that were comparable to collagen-coated sandwich hepatocyte cultures.

While many microfluidic devices have become commercially available, they offer chambers/compartments incorporating different 2D and 3D hepatocyte-like cells and NPCs granting the ability to screen drug-induced hepatotoxicity in a dynamic and complex system. For instance, the HμREL chip is manufactured with multiple interconnected cellular compartments including HPH co-cultured with NPCs, allowing fluid flow between compartments. Novik et al. treated this model, including HPH and fibroblasts, with 38 compounds composed of 19 well-characterized molecular entities, 12 hepatotoxicants, and seven non-hepatotoxic compounds, and calculated the time-based ratio by dividing the compounds TC50/Cmax value at 24 h by the TC50/Cmax values measured after dosing for 7 or 13 days [195]. The co-culture chip model detected the 24-hour-to-7/13-day time-based toxicity ratio as 33% true positives at 7 and 13 days, 67% false negatives at 7 and 13 days, 86% and 71% true negatives at 7 and 13 days, respectively, and 14% and 29% false positives at 7 and 13 days, respectively. Using 100 × human Cmax as the threshold for hepatotoxicity, the model correctly identified 10 out of the 12 compounds associated with iDILI. While the sensitivity warrants improvement, such models might serve as a useful adjunct to traditional spheroid models and other microfluidic systems in the prediction of DILI-associated drugs.

Another example is LiverChip, marketed as a two-compartment device providing HPH on an upper channel sandwiched with an extracellular matrix and an endothelial layer as the lower channel [185]. This model permits KCs to be added to the lower channel without compromising the function of the hepatocytes or requiring extra media volume. Recently, Ewart et al. evaluated a blind set of 27 known hepatotoxic and non-hepatotoxic drugs in a LiverChip model seeding HPH and LSECs in a 1:1 ratio [196]. The LiverChip correctly identified toxicity in 12 out of 15 hepatotoxic drugs and did not falsely identify any nonhepatotoxic drugs, suggesting the LiverChip model holds the potential to identify DILI drugs with a high degree of sensitivity and specificity.

Although organ-on-a-chip models remain in their infancy, other microfluidic platforms that capitalize on a two-channel system have been used to recapitulate 3D liver sinusoid models. For example, a two-channel model with a separated hepatocyte channel, composed of HepaRG cells co-cultured with stellate cells, and a vascular layer, composed of endothelial cells and tissue macrophages, has been developed for future toxicological screening [197]. Major limitations to the broader use of liver-on-a-chip systems are the cost associated with it, the lack of standardized protocols, and the fact that the extraction of cells from specific sections of the culture area can be challenging. Research is ongoing and advancements are being made with a plethora of liver chip models; however, these chip models have not been comprehensively characterized with regard to the molecular phenotypes, and their predictive power for DILI remains ill-defined.

### 3.3. Challenges Associated with Establishing Hepatocyte–NPC Co-Culture Models for DILI

Advancements in spheroid and microfluidic models have permitted the early-stage preclinical assessment of hepatotoxicity with an emphasis on cytotoxicity endpoints. While no single in vitro model can currently recapitulate all human hepatotoxic mechanisms, efforts are under way to combine multiple models to effectively increase the biological complexity and risk assessment for iDILI prediction. Currently, the majority of these iDILI studies are limited in the number of drugs screened, with only 11 studies screening more than 10 compounds [131,155,195,198,199,200,201,202,203,204]. Within these 11 studies, four studies use hepatoma cell lines, and three others use human iPSC-derived hepatocyte-like cells. The rest of the studies (4/11) utilized HPH. Additionally, the majority of the studies that screened compound libraries with >10 compounds used liver cells in the spheroid format, and only one of the models included NPC.

Specifically, human-derived cells, namely, HPH, contribute more physiologically relevant in vitro evidence, capturing population heterogeneity and retaining metabolic capacity. Conversely, simple immortalized cell lines enable high-throughput application, testing a wide variety of compounds and investigating a plethora of conditions. More complex systems, including 3D-based models, recapitulate cytotoxicity, biotransformation, and drug accumulation depending on the maintenance of 3D architecture, shear force, and concentration gradients of oxygen, nutrients, and drugs/metabolites. The combination of hepatocyte-like cells and NPCs, especially those with immune cells, adds the ability to investigate additional biological responses that contribute to DILI, such as cholestasis, steatosis, and inflammation [205].

Additionally, establishing these NPC–liver-co-culture models as DILI prediction studies is complicated by the unclear disease-related endpoints, limited understanding of iDILI mechanism, potential for substrate-specific mechanisms, and the historic focus on cytotoxicity (Table 2). With exceptions, hepatotoxicity studies emphasize cytotoxicity and do not always evaluate specific clinical manifestation or specific hepatotoxic markers for iDILI, such as disruption of tissue repair, immune response, mitochondrial dysfunction, alterations in bile salt transportation, lipid accumulation, or accumulation of metabolites [206]. However, with advancements in models, hepatocyte functionality, morphology, and biomarkers have been utilized to comprehensively assess the hepatotoxicity.

Furthermore, the drug response can vary notably in different species with hepatobiliary systems exhibiting pronounced species differences [207]. For example, species difference has been mimicked in human vs. rodent liver spheroids where human and mouse hepatocyte spheroids detected acetaminophen-induced toxicity not observed in rat and primate spheroids [201]. While a safety assessment still requires the application of animals, the sensitive species should be taken into consideration to accurately assess hepatotoxicity that resembles the toxicity and clinical manifestation observed in humans.

## 4. Conclusions and Future Perspective

This review has described a number of complex in vitro systems that are being developed to better help predict iDILI, which remains a difficult toxicity to evaluate non-clinically. There have been great strides made in the advancement of these systems, but there are currently no in vitro systems available to predict iDILI accurately and consistently for drug development purposes. While these complex in vitro systems are extremely useful during drug discovery and in order to de-risk candidate drugs, their usefulness in drug development remains to be evaluated. Additionally, iDILI is a multifaceted toxicity with numerous underlying mechanisms, mediated by various cell types that work in concert with each other, and which we have described in this review. This adds to the complexity of identifying appropriate in vitro methods that can be used as predictive tools in drug development. To improve iDILI prediction models, evidence-based training compounds are used to gain knowledge about the challenges existing with the identification and detection of iDILI-associated compounds within an in vitro or in vivo model. Drugs that make it to market and are later withdrawn due to iDILI provide a tool to test the predictive accuracy of models. The mechanism by which drugs cause idiosyncratic liver injury remains poorly defined. Subsequently, the detection of hepatotoxicity potential during preclinical assessment and clinical development is difficult. The inclusion of NPCs with hepatocytes in co-culture models provides an opportunity to better study iDILI, because it allows improved cell–cell interactions and contains an immune-responsive system. An overall goal for these in vitro models is to predict drugs with hepatotoxic potential and, thus, recapitulate a complex biological process that is multi-step and multicellular; however, these mechanistic strategies rely heavily on our present, but limited, knowledge of the iDILI mechanisms. 

Recently, major progress has been achieved towards developing physiologically relevant hepatic in vitro models to support the detection and prediction of iDILI, namely, through the application of complex culture techniques and microfluidic platforms. Despite the tremendous variability of available models and cell sources, co-culture strategies improve hepatocyte function and sustainability, while also allowing recapitulation of multicellular mechanisms involved in idiosyncratic liver toxicity. Whereas an abundance of evidence supports the immune-mediated mechanisms, there are exceptions that make extrapolating findings to clinical observations challenging. Immune-mediated toxicity can occur over a wide timeframe and in different cell types. Thus, the mechanism is both multicellular, multi-stage, and an intricate signaling process involving antigen-presenting cells, T cells, and the liver. Mimicking this interaction in vitro, including the complexity of signaling between KCs, DCs, stellate cells, and T cells, and recapitulating the 3D architecture of those systems, will enhance the understanding of the mechanisms underlying iDILI. 

However, the added value of co-culture systems has been demonstrated by diclofenac and troglitazone that suggests 3D and co-culture models evoke higher sensitivity in detecting cytotoxicity. Yet, a paradigm shift from cytotoxicity evaluation to a detailed and comprehensive multi-mechanistic evaluation is required to properly extrapolate in vitro findings and create further in vitro models with physiological relevance. Multiple factors may be involved in mechanisms of iDILI that are particularly dependent on the drug eliciting the liver injury. While advancements in these models have provided a promising opportunity to investigate the pathophysiology behind hepatotoxicity, a major difficulty remains with integrating an immune component, the standardization of protocols, the reproducibility of the detection of positive and negative controls across models, and reproducing the novel drug-specific mechanisms of iDILI. Importantly, looking at large drug screens with multiple endpoints will be critical to determine the applicability of each model. Additional development of these models, namely, the long-term co-cultures and microfluidic chips, will greatly enrich evidence of underlying iDILI mechanisms, drug toxicology, reproducibility, and inter-laboratory variation, further predicting iDILI during preclinical assessment.

## Figures and Tables

**Figure 1 cells-12-01476-f001:**
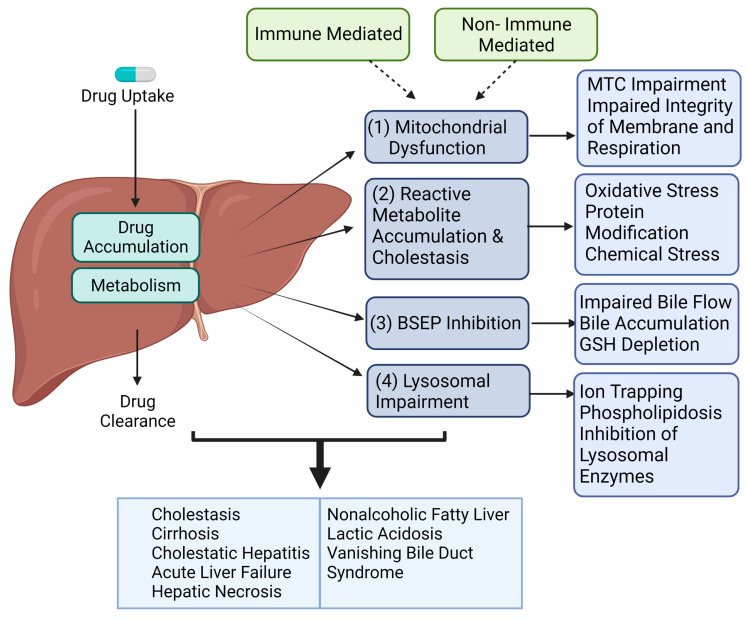
Schematic of potential mechanisms of iDILI. Immune and non-immune mechanisms are both critical, and contribute to (1) mitochondrial dysfunction, (2) reactive metabolite accumulation and cholestasis, (3) BSEP inhibition, and (4) lysosomal impairment which contributes to the development of cirrhosis, acute liver failure, hepatic necrosis, nonalcoholic fatty liver, vanishing bile duct syndrome, lactic acidosis, cholestasis, and cholestatic hepatitis. The schematic figures were generated using BioRender (biorender.com, accessed on 10 January 2023).

**Figure 2 cells-12-01476-f002:**
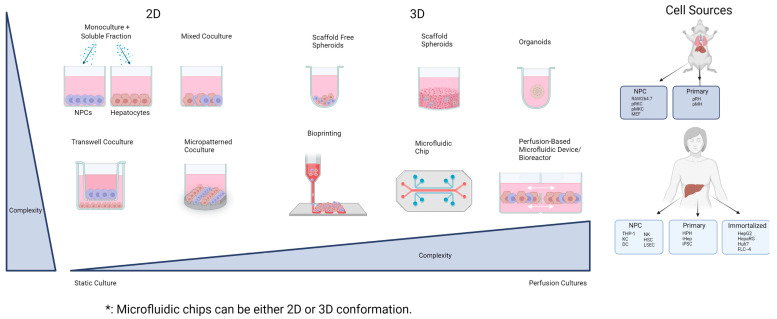
Schematic of co-culture models and cell sources. The schematic diagram illustrates static two-dimensional models such as monocultures, mixed co-cultures, transwell co-cultures, and micropatterned co-cultures and three-dimensional models such as spheroids, organoids, bioprinting, microfluidic chips, and perfusion-based microfluidic devices/bioreactors (e.g., 3D liver-on-a-chip). Models can include 2D and 3D components. It also highlights liver cell sources that are often used in these models including primary, immortalized, and NPC cells from different species.

**Table 1 cells-12-01476-t001:** Drugs withdrawn from the US market from 1996–2010.

Drugs	Date Approved	Withdrawn Date	Reason for Withdrawal	Potential Mechanism of DILI
Troglitazone	29 January 1997	21 March 2000	Liver failure	GSH depletion [42], reactive species formation [43], and BSEP inhibition [44]
Bromfenac	15 July 1997	22 June 1998	Severe hepatitis and liver failure	GSH depletion and reactive species formation [45,46]
Trovafloxacin	1997	15 June 1999	Acute hepatitis	Immune-mediated [47]
Tolcapone	25 January 1998 & 31 August 2009	November 1998	Acute liver failure	Mitochondrial dysfunction and reactive species formation [48,49]
Nefazodone	5 May 2003	14 June 2004	Acute liver failure	BSEP inhibition [50] and reactive species formation [51,52]
Sitaxentan	15 June 2007	10 December 2010	Liver injury	BSEP inhibition [53]

**Table 2 cells-12-01476-t002:** Advantages and challenges with establishing NPC–liver-co-culture.

	Model	Challenges	Advantages	References
2D	Monoculture + soluble fraction	No cell–cell interaction	Easy to culture, improved hepatic phenotype	[93,94]
	Mixed culture	Random distribution of cells	Physical cell–cell interaction	[114]
	Transwell	Heterogenic cell–cell interaction, degree of separation	Increased expression of pro-inflammatory cytokines/chemokines	[131]
	Micropattern	Cell–ECM detachment, no direct heterogenic cell–cell interaction	Phenotypic stability	[112]
3D	Spheroids & organoids	Potential for toxicity from synthetic scaffolds, increased potential of false positive for drugs	Viable for the long term and maintained stable transcriptomes/proteomes	[157]
	Bioreactors	Not amenable to high-throughput screening, hydrodynamic shear forces	Concentration nutrient gradient, maintained hepatic metabolic function	[179]
	Microfluidic liver-on-a-chip	Lack standardized protocol, not amenable to high throughput	3D architecture, layers of NPCs adjacent to hepatocytes, extracellular matrix	[190]
	Bioprinting	Not amenable to high-throughput screening, expensive	High resolution, 3D architecture, and phenotypic stability	[172]

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
