# Peer review of "Microphysiological Models for Mechanistic-Based Prediction of Idiosyncratic DILI"

_cells, 2023, doi:10.3390/cells12111476_

Round 1

Reviewer 1 Report

The current work summarizes the in vitro co-culture models for studying idiosyncratic DILI (iDILI). The authors highlighted the advancements and limitations in human-based 2D/3D multicellular models.

Comments

1.       The mechanisms for iDILI are very complicated.  The authors listed four mechanisms in the session of “2. Mechanism of DILI in Humans” and Fig 1. To be more inclusive, the authors may add information of other mechanisms in this part.

2.       The authors showed that reactive metabolites can lead to oxidative stress (Fig 1). Details for the consequences of mitochondrial dysfunction, BESP inhibition, and lysosomal impairment should also be added in Fig 1.

3.       Legend for Fig 1: “vanishing bile duct syndrome, cholestasis, and cholestatic hepatitis” have similar phenotypes as cholestatic injury. The authors may change “phenotypes” to “development” or “pathogeneses”.

4.       The authors may split the session “2.2. Reactive Metabolite Accumulation & Cholestasis” into two different sessions, because these two parts are not well-connected.

5.       References should be provided in Table 1, especially for the column of “Potential Mechanism of DILI”.

6.       More information should be provided in the legend of Fig 2 “Schematic of coculture models and cell sources”.

7.       Table 2 can be cited early in the session of “3. Overview of Hepatocyte-NPC coculture models”. In addition, the legend for Table 2 “Challenges with Establishing NPC – Liver Coculture” can be changed to “Advantages and Challenges with Establishing NPC – Liver Coculture”.

Author Response

See attached file for our responses.

Reviewer 2 Report

This review comprehensively summarizes the most recent data on mechanisms of idiosyncratic DILI development, and numerous cell models of DILI. Nevertheless, some improvements are necessary in order to make it more valuable.

Considering that numerous models were described in the manuscript, it would be more clear to the readers if the most important models (and cells used for establishing these models), along with the literature were listed in one Table.

Minor corrections regarding the quality of English are necessary. Some of the suggestions are listed below:

Line 38- the market

Line 50- have

Line 54- does

Line 75- is not

Line 101- the majority

Line 109- , and

Line 111- the majority

Line 128- the market

Line 164- the evidence

Line 195- diclofenac

Line 226- that was

Line 230- the market

Line 240- mechanisms

Line 283- the market

Line 205- while increasing

Line 501- the functional

Line 558- reprogramed

Line 606- but a promising

Line 684- Haque et al. evaluated

Reviewer 3 Report

1)    The manuscript by Stern et al reviews microphysiological models for prediction of idiosyncratic drug induced liver injury. While generally well written, the authors need to distinguish their review from a very similar one published very recently (PMID: 35024301), which essentially covers the same topic. They could probably mention updated methodologies, for example.

2)    Page 2, line 82- Please change sentence to -……..the typical start is 1-3 months after drug consumption started.

Reviewer 4 Report

This review aimed at describing the involvement / importance of inflammatory response on iDILI – and how this can be predicted. Specifically, the in vitro models (especially 3D) available to study this. Overall, the review is generally well written, but does require a “language” check.

There is also often a confusion in the distinction between DILI and iDILI throughout the methods / examples described. Often iDILI is used when in fact the data/model only reflects general hepatotox. The involvement of the immune component is only sporadically described (which does reflect the state of the field at the moment).

Throughout the manuscript, the content often appears “out of place” – eg repetitive, inclusion in wrong section, consistency across the different sections (eg including a short statement on disadvantages only mentioned for some models – as are specific hepatotox examples).

I do believe, with a significant edit, this review is important to publish.

Specific comments:

Introduction:

I would argue that DILI IS a life threatening ADR?

Flow doesn’t always make sense – sentences appear out of context (eg ADME on lines 49-51). The description of predictive DILI (even though not focus) is very weak… only mention acetaminophen!

Not sure the inclusion of NPCs is a “recent advancement” – have been included in spheroids for many years!

2. Mechanisms of DILI

Language needs to be addressed, eg “…reflecting the role of drug liver toxicity on a wide array of phenotypes“  - liver tox doesn’t have a role in these phenotypes.

Need to be clearer about nomenclature – eg dose-dependent/predictive / intrinsic….

Current mechanistic understanding (lines 89-)(and following sections) – are generic for DILI and not iDILI – please be clearer.

Non-inflammatory iDILI – mechanisms at least mention – with refs?

2.1 Immune

Line 117 – inclusion of LPS needs context

Table: last withdrawal was 2010 – maybe change title to table. I assume no further withdrawals were seen after 2010…

2.2 Reactive metabolites

Inclusion of paragraph on cell death mechanisms seems out of place under this heading.

2.4 BSEP

Old nomenclature used for transporters (OK to include, but would be good to include the new eg BSEP = ABCB11).

2.5 Lysosomal impairment

Phospholipidosis is described but not mentioned?

2.6 Regulatory considerations

Mention the FDA Modernization Act?

Hys’s Law mentioned: by nothing around other approaches such as ALT/AP ratios now being discussed in literature.

Hep-NPC CoCultures

Section is a little muddled in my opinion. The flow isn’t right? Eg., Cell lines aren’t really relevant under this section heading…

Line 336: more predictive of what?

KC/HPK ratio statement needs to be expanded – 1 sentence doesn’t tell us anything…?

Fig. 2. Careful – not all microfluidic chips are 3D – maybe need an additional group for MPS?

3.1.1: not sure that inclusion of NPCs alone means you will see adaptive immune responses (maybe innate?). Any examples?

Trovafloxacin/Acetaminophen example – what relation to actual hepatotox was observed?

3.1.12: Alb/ATP/GSH are general hepatic markers – I wouldn’t call then iDILI sensitive endpoints! Not clear if compound set included caused only iDILI or DILI?

3.2.2: difference between iPSC and adult stem cell derived? Any evidence that the derived organoid from one specific donor mimics that donor?

3.2.4: Disadvantages of perfusion-based models? Was included for other models and is critical for this review.

3.2.4.2: Haque et al: say they evaluated function importance – but what was the conclusion form this research?

Novik et al: 12 hep + 7 non hep - + ?? (29 compounds tested)? Data summary form this paper is also a little confusing!

Why not mention Emulate (ie LiverChip) when Hurel is mentioned? Include the recent Emulate DILImp0rediction paper from Ewart et al (Dec 2022)

The disadvantages of OOC models not discussed.

3.3: Nice summary – probably could be expanded.

Not sure DILI prediction is hampered by lack of disease-related endpoints (different question usually…). Also don’t agree that most endpoints in all these advanced models focus solely on cytotox. Many other endpoints have been reported.

The paragraph on animal models is important, but appears a little unstructured – and should be more general (eg don’t need the acetaminophen story …).  Not sure focusing on the human metabolites is relevant in this review.

Conclusion / Future:

Most of the models discussed are more generally focusing on DILI prediction - and not iDILI. Maybe need to be clearer in intro that majority of models discussed are focusing on FILI per se – and not iDILI.

No mention of models using APC, T cells in the manuscript? Are these being assessed in any of the models discussed?

Round 2

Reviewer 1 Report

None

Reviewer 3 Report

The authors have responded satisfactorily to the comments.